# Bibliometric Analysis of Research on Exercise Intervention for Cancer-Related Cognitive Impairments

**DOI:** 10.3390/healthcare12191975

**Published:** 2024-10-03

**Authors:** Yuwei Shen, Ningsheng Xu, Tingting Yu, Jianan Li

**Affiliations:** 1School of Rehabilitation Medicine, Nanjing Medical University, Nanjing 210029, China; shenyuweisyw@stu.njmu.edu.cn (Y.S.);; 2Department of Rehabilitation Medicine, The First Affiliated Hospital of Nanjing Medical University, Nanjing 210029, China

**Keywords:** bibliometric analysis, cancer, cognitive impairments, exercise, physical activity

## Abstract

**Introduction:** Cancer treatments frequently lead to cognitive impairments, affecting a substantial global population. Among various approaches, exercise has emerged as a promising strategy for rehabilitation. However, a comprehensive bibliometric analysis of research in this field is lacking. **Methods:** We conducted a bibliometric analysis of 10,345 articles sourced from the Web of Science database using the R package “bibliometrix”. Our analysis examined publication trends, leading countries, journals, authors, institutions, keywords, and prevalent themes. **Results:** Over the past two decades, research on exercise interventions for cancer-related cognitive impairments (CRCI) has advanced significantly. Nonetheless, challenges persist in elucidating underlying mechanisms, developing innovative strategies, and creating effective tools. **Conclusions:** The number of publications notably increased from 1998 to 2023, although there has been a recent decline in citations. The United States (US) leads in both publications and citations, while China is showing increasing influence. Using Lotka’s Law in our bibliometric analysis, we identified 58 key authors in the field of exercise interventions for CRCI. Leading institutions such as the University of Toronto and Duke University are at the forefront of this research. Although the *Journal of Clinical Oncology* has fewer publications, it remains influential. Current research focuses on exercise interventions to enhance the quality of life for cancer patients, with particular emphasis on cognitive rehabilitation in breast cancer and the challenges faced by survivors. Future research should delve deeper into intervention mechanisms, behavioral strategies, telemedicine, and precise cognitive assessment tools.

## 1. Introduction

Cancer remains a significant global public health challenge, with millions of new cases reported annually, posing a grave threat to human life quality and life expectancy [1]. Despite notable progress in improving the five-year survival rate of cancer patients through chemotherapy, which has risen from approximately 10% to about 50% [2], the associated side effects, particularly cognitive dysfunction, have emerged as a critical factor affecting patients’ quality of life and rehabilitation progress [3]. Chemotherapy-induced cognitive dysfunction, often metaphorically referred to as “chemo brain” or “chemo fog”, manifests as difficulties in learning, distracted attention, and memory decline, significantly impacting patients’ daily activities and social functioning [4]. Clinical data indicate that the incidence of self-reported cognitive impairments ranges from 17% to 78% [5], while neurocognitive testing confirms cognitive deficits in approximately 33% of breast cancer patients undergoing chemotherapy [6].

Although the negative impact of cognitive dysfunction on cancer patients’ recovery cannot be overlooked, current management strategies remain limited. The efficacy of pharmacological interventions is modest [7], while the effectiveness of cognitive behavioral therapy and other non-pharmacological approaches awaits further validation [8,9]. In this context, exercise, as a safe, economical, and easily implemented intervention, has garnered significant interest from researchers and clinicians. Exercise not only brings significant long-term benefits to the physical health of cancer patients, such as improving cardiopulmonary function [10,11], enhancing muscle strength [12,13], alleviating cancer-related fatigue [14,15], reducing cancer pain [16,17], and mitigating chemotherapy-induced peripheral neuropathy [18,19], but also actively promotes their mental health, alleviates anxiety and depression [20,21], and exerts a positive impact on the overall recovery process. However, despite the growing research on exercise interventions in cancer-related cognitive impairment (CRCI), a systematic quantitative analysis of the overall research status, hot topics, and cutting-edge trends in this field remains insufficient.

Bibliometrics is an interdisciplinary research method that integrates mathematics, statistics, and informatics. Using scientific data quantification techniques, it enables a comprehensive and structured analysis and interpretation of literature in specific research areas. Combined with visualization tools, bibliometrics can dynamically and intuitively showcase the evolutionary trajectory and knowledge framework of research fields, revealing their current research status. This provides precise and detailed informational references for future research planning and strategy formulation [22]. This methodology has been widely applied across various disciplines, including medicine, management, informatics, and biology [23,24,25,26]. Through its unique combination of quantitative and qualitative analysis, bibliometrics can deeply reveal the contribution levels of different countries, journals, authors, and institutions in specific research areas. It clarifies commonly used research methods and keywords and constructs an international collaboration network map for the field [27]. Compared to traditional review methods, bibliometrics exhibits higher precision and completeness in presenting the development context and trends of disciplines.

Given the aforementioned background, this study selects relevant literature from the Web of Science Core Collection (WoSCC) to construct a research sample. Utilizing the “bibliometrix” package in R language, combined with bibliometrics and visualization analysis techniques, the aim is to conduct a deep exploration of the current status of research on exercise interventions in CRCI. By identifying research hotspots and predicting development trends, this study aspires to fill the gaps in the depth and breadth of quantification in existing reviews. It provides valuable references and insights into the role of exercise interventions in cancer rehabilitation, especially cognitive function recovery, thereby promoting the deepening and expansion of related research.

## 2. Materials and Methods

### 2.1. Data Source and Literature Search Strategy

This study selects Web of Science as the core database, with the data retrieval cutoff date set to 12 March 2024. The Web of Science Core Collection is a major global source of academic information, encompassing over 21,800 authoritative and high-impact scholarly journals across a range of fields, including natural sciences, engineering and technology, biomedicine, social sciences, arts, and humanities [28]. Although Google Scholar offers broader coverage, it also includes non-academic and low-impact literature, necessitating additional processing to filter out high-quality academic materials. In contrast, while Web of Science may not match Google Scholar in terms of citation coverage, its performance in terms of literature quality and citation completeness has surpassed that of Scopus and Google Scholar since 2021 [29].

Based on Liu’s recommendations in *Scientometrics* [30], our Web of Science Core Collection data retrieval focuses on the following key subsets: the Science Citation Index Expanded (SCI-E), which dates back to 1900 and is the most authoritative natural science citation database globally, encompassing a large number of high-impact natural science journals; the Social Science Citation Index (SSCI), also starting from 1900, which is the most authoritative database in the social sciences, providing extensive academic resources for social science researchers; the Arts and Humanities Citation Index (A&HCI), which includes journals dating back to 1975 and covers a broad range of academic journals in the arts and humanities; the Conference Proceedings Citation Index (CPCI), including CPCI-S (Science) and CPCI-SSH (Social Science and Humanities), which primarily indexes conference proceedings across various disciplines, typically covering the years from 1990 to the present; and the Book Citation Index (BkCI), focusing on citations of scientific books and monographs, usually covering the years from 2005 to the present.

After consulting with experienced literature search specialists, all authors agreed on the search strategy, as outlined in Table 1. For ease of further analysis, only journal articles and reviews written in English were included, and complete records and reference citations from relevant publications were extracted and saved in plain text format for subsequent research.

### 2.2. Statistical Analysis

This study utilized R version 4.3.3 [31] and Bibliometrix R package version 4.0.0 [32] as the software tools for conducting the bibliometric analysis. Biblioshiny is an open-source online visualization tool based on the Rstudio platform, excelling in bibliometric statistical analysis, complex network model construction, and knowledge map drawing [32]. Specifically, we covered statistics on the number of publications and citations over the years, a comparative analysis of scientific research output among countries, the construction of international cooperation networks, the examination of high-yielding authors’ academic contributions and their cooperation networks, the investigation of research institutions’ contributions and cooperation networks, statistics of high-impact journals, and the citation analysis of key papers. Furthermore, we delved into research keywords and themes, utilizing visualization techniques to intuitively showcase the core development trends, evolution of research hotspots over the years, inter-thematic correlations, and mutual influences in this field. Through this series of comprehensive analyses, we strove to reveal the development course, current research status, and future trends of this field.

## 3. Results

### 3.1. Overall Publication Status

A preliminary search yielded a total of 11,455 relevant articles. During the screening process, 426 non-English documents and 684 items not meeting the inclusion criteria (including conference abstracts, journal reviews, and research briefs) were excluded. After removing duplicates, 10,345 articles were finalized for the study, comprising 8333 articles and 2012 review articles (see Figure 1).

Publication trends serve as a direct reflection of the research level and popularity within a specific discipline or field. Research on the effects of exercise intervention on CRCI spans from 1998 to 2023, with annual publication trends depicted in Figure 2A. Since the publication of the pioneering paper “Cancer Pain: A Provocation of Emotional, Social, and Existential Distress” [33] in 1998, interest in this area has steadily increased. From 1998 to 2013, the number of publications exhibited a consistent upward trend, indicating a period of stable growth. However, there was a brief decline in publication numbers from 2013 to 2014. Since 2014, the volume of related articles has surged, marking a phase of rapid development. It is important to note that the growth in publications may also be influenced by the inclusion of new sub-datasets and the expansion of Web of Science, as suggested by researchers such as Liu [34]. These factors should be considered when interpreting the trends. Overall, despite some fluctuations, research on exercise interventions for CRCI has demonstrated a consistently upward trajectory.

Figure 2B illustrates the trend in the average annual citation frequency from 1998 to 2022. In the initial period (1998–2002), the citation frequency gradually declined from a relatively high level, but then remained relatively stable in the subsequent years (2002–2006). From 2006, the citation frequency experienced a period of fluctuation, with a slight increase (2006–2010) followed by a plateau (2010–2014). However, since 2014, the citation frequency has shown a downward trend again, particularly with a sharp decrease after 2018, reaching a very low level by 2022. Overall, the findings indicate a long-term decline in citation frequency, particularly evident in the last decade. Although research in this field has maintained high academic value and experienced citation peaks during specific time periods, data from 2019 to 2023 suggest a yearly decrease in citation frequency for literature in this field, potentially indicating that research in this area has entered a relatively mature stage, with a corresponding reduction in the need to cite earlier research outcomes.

### 3.2. Analysis of Countries and Source Journals

#### 3.2.1. Analysis of Highly Productive Countries

In the field of study, the top ten most prolific countries, based on the affiliation of corresponding authors, are the United States (US), China, the United Kingdom (UK), Australia, Canada, Germany, the Netherlands, Italy, France, and Japan, respectively. Among these nations, developed countries predominate, with China being the sole non-developed country represented (refer to Table 2 and Figure 3A). In terms of citation frequency, the top ten countries exhibit slight variations from the aforementioned ranking, specifically, the US, the UK, Canada, Australia, Germany, the Netherlands, China, Italy, France, and Denmark. Detailed data reveal that the US has garnered 180,253 citations for 3521 publications, whereas China has accrued 13,799 citations for 965 articles. This comparative analysis underscores the research output and global influence of various countries in the domain of exercise intervention for CRCI.

As evident from Figure 3A, the United States leads globally in terms of both solo and collaborative publications pertaining to exercise interventions for CRCI, significantly outpacing other nations. Notably, despite China’s second-place ranking in terms of total publication volume, surpassing countries such as the United Kingdom, Australia, and Canada, the latter three nations demonstrate higher levels of activity in terms of international collaborations when compared to China.

Based upon further examination of international collaborations, Figure 3B illustrates the total volume of published papers by each country through variations in map shading, while the thickness of the red lines provides a visual representation of the closeness of cooperation between nations. The data indicate that the US is at the forefront of international collaborations in this field, with particularly strong ties to Canada (310 collaborations), followed by the UK (172 collaborations) and Australia (168 collaborations). In the context of China’s international collaborations, the US emerges as the primary partner (126 collaborations), trailed by Australia (43 collaborations) and the UK (31 collaborations). These statistics provide compelling evidence for the increasingly prominent trend of globalization and cooperation in the research realm of exercise interventions for CRCI.

#### 3.2.2. Analysis of Highly Productive Journals

Based on the data presented in Table 2, it is evident that, in the field of exercise interventions for CRCI research, the journals *Supportive Care in Cancer* and *Psycho-oncology* have published a significantly higher number of relevant papers compared to other similar journals. Although the *Journal of Clinical Oncology* has a lower publication volume in this field compared to the aforementioned journals, its total citation count is higher, indicating stronger academic influence and literature value.

Upon further analysis of historical data, it is observed that the publication volume of various journals generally shows a trend of increasing year by year. It is worth noting that prior to 2021, *Psycho-oncology* maintained a higher publication record than *Supportive Care in Cancer* for consecutive years. However, since 2021, *Supportive Care in Cancer* has achieved a reversal in publication volume in this field, surpassing *Psycho-oncology* and becoming a more active journal in the field (see Figure 4).

### 3.3. Author and Institution Analysis

#### 3.3.1. Author Productivity through Lotka’s Law

Lotka’s Law describes the mathematical relationship between the number of authors publishing a certain number of papers and their distribution within a population. Specifically, in bibliometrics, the ratio of authors who have published n papers to those who have published only one paper follows an approximate ratio of 1:1/n^2^. By applying Lotka’s Law in productivity analysis, we can effectively identify the “core authors” in a research field—those who produce a relatively high and consistent output [35]. As illustrated in Figure 5, in the domain of exercise intervention for CRCI, there are approximately 58 core authors who have published at least 10 papers, accounting for 0.1% of all authors. This group contributes significantly and concentratedly to the field. On the other hand, 36,800 authors, constituting 81.5% of the total, have only published one research paper each, forming the so-called “non-continuous research zone” or “low-frequency output zone”.

#### 3.3.2. Analysis of Highly Productive Institutions

As depicted in Figure 6A, there exist significant disparities in research output among institutions in the field of exercise intervention for CRCI. The University of Toronto leads the pack with 538 publications, closely followed by Duke University with 438 papers. The University of California, San Francisco, and the University of California, Los Angeles, have produced 424 and 423 papers, respectively. Additionally, the University of Sydney has contributed 420 studies to this field. It is worth noting that the University of Texas MD Anderson Cancer Center, as a prestigious cancer research center, has published 370 relevant papers (refer to Table 3 for details).

The network structure further reveals the formation of three tight clusters among these research institutions (see Figure 6B). Notably, the University of Sydney, the University of Toronto, and the University of Michigan exhibit high betweenness centrality, indicating their critical role in connecting various components of the network and facilitating information flow. Consequently, these three universities are highly likely to serve as central nodes for academic exchanges and play a pivotal role in bridging interdisciplinary collaboration and knowledge dissemination on a global scale.

### 3.4. Citation Analysis of Papers

Among the top 10 most-cited papers in the field of exercise intervention for CRCI listed in Table 4, the paper published by Kerry S Courneya et al. in the *Journal of Clinical Oncology* in 2017 stands out with the highest citation count of 117, occupying the top spot. This paper delves into the effects of aerobic and resistance exercise on breast cancer patients undergoing adjuvant chemotherapy through a multi-center randomized controlled trial. It is worth mentioning that Kerry S Courneya is one of the most prolific authors in this field, and his research outputs have garnered widespread recognition and citations [36]. Therefore, for scholars new to this field, this paper is undoubtedly a classic and essential reading, providing insights into the fundamentals of exercise interventions in CRCI.

Furthermore, among these ten highly cited papers, the *Journal of Clinical Oncology* contributed half of them. In contrast, despite having the highest number of publications in this field, the journals *Psycho-Oncology* and *Supportive Care in Cancer* lack highly cited papers. This suggests that although these two journals may have an advantage in terms of the overall number of publications, they are slightly less impactful in producing high-quality research compared to the *Journal of Clinical Oncology*.

### 3.5. Keyword Analysis

#### 3.5.1. Keyword Frequency Distribution

Keywords provide a concise summary of the content and themes of academic papers, directly reflecting research hotspots in the current field. Setting the keywords as “author-selected keywords”, Figure 7 visualizes the frequency of the top 50 keywords in the form of cube sizes. As shown, the top 50 keywords with a frequency exceeding 2% account for 56% of all keywords, including high-frequency terms such as cancer (*n* = 1531, 12%), quality of life (*n* = 1096, 9%), breast cancer (*n* = 844, 7%), exercise (*n* = 634, 5%), physical activity (*n* = 617, 5%), oncology (*n* = 576, 5%), depression (*n* = 533, 4%), fatigue (*n* = 388, 3%), chemotherapy (*n* = 348, 3%) and anxiety (*n* = 333, 3%). These high-frequency keywords represent the core topics of current research.

#### 3.5.2. Trend Analysis of Keyword Hotspots over the Years

The evolution of research hotspots in exercise intervention for CRCI is reflected in the changing keywords over the years (see Figure 8). In 2021, the focus was on “inflammation”, “physical function”, and “statement”. In 2022, research shifted towards “behavior change interventions”, “cell cycle arrest”, and “postoperative pain”. By 2023, key areas of interest included “telehealth”, “COVID-19”, and “cognitive assessment toolkits”. Future research is expected to concentrate on analyzing specific mechanisms, optimizing strategies with advanced technologies, developing precise cognitive assessment tools, and addressing overall health status and adaptive strategies in special circumstances.

#### 3.5.3. Thematic Map

The thematic map (refer to Figure 9) serves as a visual tool based on keyword-driven conceptual structures, aiming to delve into the current research landscape and interconnections within the field of exercise interventions in CRCI. This map is meticulously divided into four quadrants based on two core indicators: relevance and developmental maturity.

The themes in the first quadrant, designated as “mainstream” or “dynamic” due to their high research intensity and established frameworks, include “physical activity”, “palliative care”, “needs”, and “aerobic exercise”. These themes represent pivotal areas of current investigation and are likely to drive future advancements.

Themes in the second quadrant, such as “mechanisms”, “inflammation”, and “tumor necrosis factor”, are on a positive development trajectory but exhibit limited integration with mainstream research, suggesting potential for future innovation.

The themes in the third quadrant, including “brain”, “risk factors”, and “stress”, are either in the nascent stages of research or being marginalized, indicating that they may evolve into emerging fields or warrant reevaluation.

The themes in the fourth quadrant, such as “cancer”, “quality of life”, “randomized controlled trials”, and “breast cancer”, are essential for understanding specific domains and have established research consensuses, though they show developmental lag.

## 4. Discussion

### 4.1. General Research Directions

This study employed bibliometric analysis to delve into 11,029 academic publications spanning from 1998 to 2023, focusing on the intersection of exercise or physical activity and CRCI. These publications were co-authored by 45,029 authors from 10,812 research institutions across 91 countries/regions, covering 2246 academic journals. Through comprehensive data analysis, we uncovered the current state and emerging trends in this research field.

Examining the temporal dimension, the number of research papers in this field exhibited a steady increase from 1998 to 2023, with significant growth since 2014. This trend indicates a rising global interest in this research area. However, it is worth noting that citation frequencies have declined in recent years, possibly suggesting that the field is maturing, and new publications find it more challenging to achieve high citations in a short period.

On a national level, the US stands out globally with its substantial number of publications and high citation rates, aligning with the findings from bibliometric analyses in the field of physical activity and cognition [37]. As a developing country, China has also made remarkable progress in this field. Indeed, China has played a pivotal role in the global scientific community, especially in recent years, significantly increasing its contribution to scientific publications. Examples include Yafeng Wang’s “Association of the ‘Weekend Warrior’ and Other Leisure-time Physical Activity Patterns with All-Cause and Cause-Specific Mortality: A Nationwide Cohort Study” [38] and Chuanmei Zhu’s “Exercise in Cancer Prevention and Anticancer Therapy: Efficacy, Molecular Mechanisms, and Clinical Information [39]”. However, there is still room for enhancing international collaboration. The international cooperation network indicates that while the US continues to engage in independent research, it also collaborates closely with other developed countries such as Canada and the UK.

In terms of academic journal influence, *Supportive Care in Cancer* and *Psycho-oncology* boast a considerable volume of publications. However, the *Journal of Clinical Oncology* demonstrates stronger academic influence due to its high total citation count. In recent years, *Supportive Care in Cancer* has emerged as a prominent platform for publishing research in this field.

In terms of author contributions, applying Lotka’s Law allowed us to identify 58 core authors who have each published at least 10 papers, making significant contributions to the advancement of the field. At the institutional level, leading research centers include the University of Toronto, Duke University, the University of California campuses in San Francisco and Los Angeles, the University of Sydney, and the University of Texas MD Anderson Cancer Center. Notably, the University of Sydney, the University of Toronto, and the University of Michigan play pivotal roles in global academic exchanges due to their outstanding international collaboration and knowledge dissemination capabilities.

By analyzing highly cited papers, we found that Kerry S Courneya’s team’s research on exercise intervention for breast cancer patients, published in the *Journal of Clinical Oncology* has had a profound impact. This paper serves as an excellent starting point and essential reading for newcomers to the field, helping them quickly grasp the current research status, methodologies, and key findings in the area of exercise intervention for CRCI.

### 4.2. Hotspots and Frontiers

In 2021, keywords like “inflammation”, “physical function”, and “statement” marked a shift towards exploring the role of inflammation in the pathogenesis of CRCI [40] and recognizing the importance of physical function in evaluating the effects of exercise interventions. Statements or consensus during this period likely established a foundation for a shared understanding of core issues in the field, guiding future research directions [41].

By 2022, the focus had shifted to “behavior change interventions”, “cell cycle arrest”, and “postoperative pain”. These keywords reflect a deeper exploration of the effects of exercise on behavioral and psychological aspects of cancer patients, as well as its direct inhibitory effect on cancer cell proliferation [42]. Additionally, consideration of postoperative pain highlights the potential role of exercise interventions in cancer rehabilitation therapy [43].

In 2023, research hotspots such as “telehealth”, “COVID-19”, and “cognitive assessment toolkits” underscored the significance of integrating technology, responding to emerging infectious diseases, and precise assessment in CRCI research. Rapid advancements in technology have enhanced the application of telemedicine in exercise intervention strategies, attracting widespread attention [44,45]. Examples include Christelle Schofield’s practical research on a 12-week remote supervised resistance exercise program for ovarian cancer survivors who have completed first-line treatment [44], as well as Alexia Piché et al.’s study on the practical feasibility of a multimodal remote prehabilitation program in cancer care [45]. The impact of the COVID-19 pandemic on cancer patients has led researchers to reflect on maintaining exercise habits and mitigating cognitive impairment under these special circumstances. Importantly, there is a commitment to developing more accurate cognitive assessment tools to precisely measure improvements in patients’ cognitive functions through exercise interventions [46,47].

Examining the distribution of keyword frequencies reveals the evolution of research and emerging trends in exercise interventions for cognitive impairment related to CRCI. (1) Improving quality of life is a central theme in current research. The high frequency of the keyword “quality-of-life” indicates that significant attention is being paid to utilizing exercise interventions to enhance the quality of life of cancer patients and survivors [48,49]. (2) Research on exercise interventions for breast cancer is particularly prominent among various cancer studies. The dominance of the keyword “breast-cancer” suggests extensive and in-depth research into exercise intervention strategies for breast cancer patients during their rehabilitation process [50,51]. Possible reasons for this focus include the large population of breast cancer patients, disease characteristics, and relatively high long-term survival rates, which necessitate efforts to improve cognitive function and quality of life for this group. (3) Most studies target cancer survivors as their primary subjects. The frequent occurrence of the keyword “survivors” confirms this trend, indicating a growing emphasis on how exercise interventions can help individuals who have overcome cancer maintain or restore cognitive abilities during the post-treatment phase, integrate better into social life, and resume normal work activities [52,53,54]. (4) Other high-frequency keywords such as “physical-activity” and “exercise” highlight that the design and application of exercise programs form the foundation of many research projects. These studies not only assess the effectiveness of exercise interventions, but also explore the specific impacts of various types, intensities, and durations of exercise on improving cognitive impairments in survivors of different cancer types [55,56,57]. The widespread presence of the term “cancer” underscores the broad applicability of research, indicating that while breast cancer research is predominant, exercise interventions hold significant potential for cognitive recovery across various cancer treatments [58,59].

Overall, analyzing keywords’ frequency and co-occurrence reveals the progression of research and emerging trends in exercise interventions for CRCI. Initially, the focus was on improving the quality of life for cancer patients and survivors [48], as indicated by the prevalence of the term “quality-of-life”. Over time, research has shifted towards specific cancer types, such as breast cancer [57], and more targeted populations like cancer survivors [60]. This shift reflects a growing interest in the effects of exercise interventions on specific cancer types and groups. Additionally, new research hotspots are emerging, including the mechanisms and effects of exercise interventions, such as behavior change [61,62] and cell cycle impacts [63]. Technological advancements, such as telemedicine [44,45] and new cognitive assessment tools [46,47], have also enhanced the accessibility and precision of exercise intervention studies.

### 4.3. Existing Limitations

While this study has provided valuable insights, it is not without certain limitations. Firstly, citation bias, a prevalent issue in scientific research, could affect the study’s findings, as citations may be influenced by the nature and direction of the results, potentially impacted by factors such as self-citations, authorship, and journal impact factors [64]. For example, studies with negative outcomes are often cited less frequently. Secondly, there may be a bias against novel research within the scientific community, as newer articles typically require more time to accumulate significant citations [65]. Additionally, although Web of Science offers a comprehensive resource for bibliometric analysis, it has its limitations, including language biases, data lag, subjective quality evaluation, and challenges in cross-disciplinary analysis [66,67,68]. Information availability for older literature, particularly before 1990, is low for abstracts, author keywords, and keywords plus fields, mainly because WoSCC did not systematically collect this information from publishers early on or because it was not provided in the older literature itself. Research indicates that the systematic recording of these fields began in the 1990s [69]. Furthermore, the various sub-databases of WoSCC (such as SCI, SSCI, and A&HCI) have different coverage and time spans, and different institutional subscribers may have access to different subsets of data, which could lead to inconsistent search results among users. Moreover, certain sub-databases, like the Conference Proceedings Citation Index, might have started collecting data extensively at a specific point, affecting the literature growth for particular years [34]. Lastly, limiting the study to English-language publications might exclude important research from non-English-speaking regions, thereby somewhat narrowing the global diversity perspective.

## 5. Conclusions

In summary, significant progress has been made in the study of exercise interventions for CRCI over the past two decades. Nevertheless, further understanding of the intervention mechanisms is still warranted in future research endeavors. We propose the integration of emerging technologies to innovate and optimize intervention strategies and assessment tools. Meanwhile, strengthening international collaboration can jointly facilitate sustained progress and development in this field. This study employs bibliometric analysis to delve into the current research status and future trends, aiming to provide some reference and insight for subsequent in-depth studies and field expansion.

## Figures and Tables

**Figure 1 healthcare-12-01975-f001:**
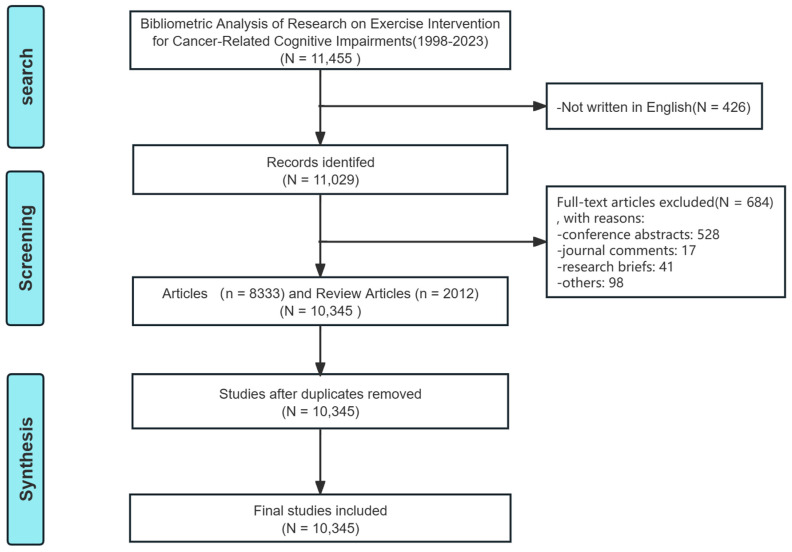
Flow diagram of the literature selection process in this study.

**Figure 2 healthcare-12-01975-f002:**
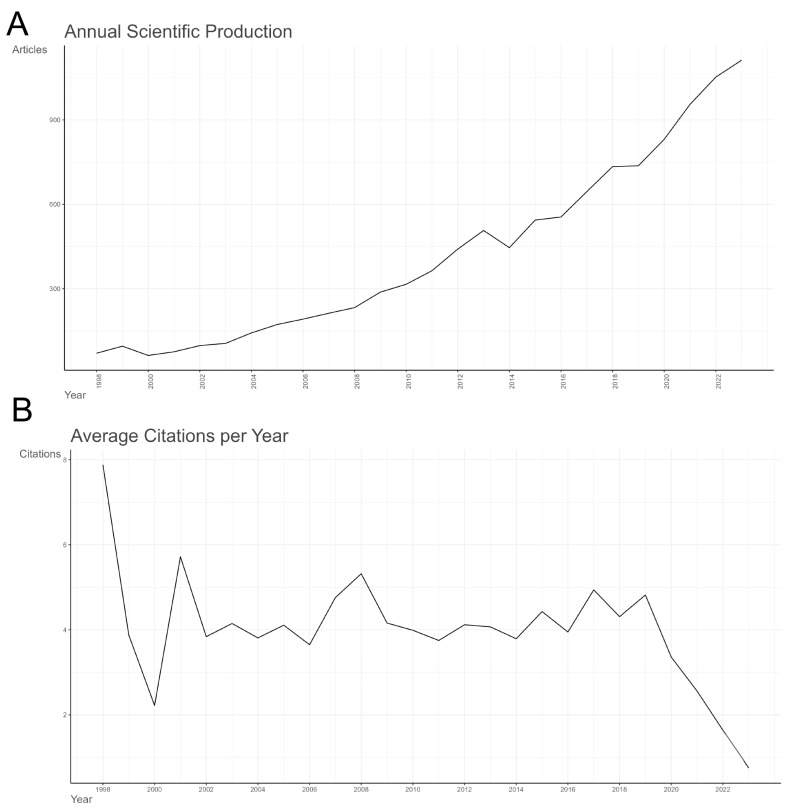
Overall publication situation. Overall trend of publications (**A**); trend analysis chart of citation frequency over the years (**B**).

**Figure 3 healthcare-12-01975-f003:**
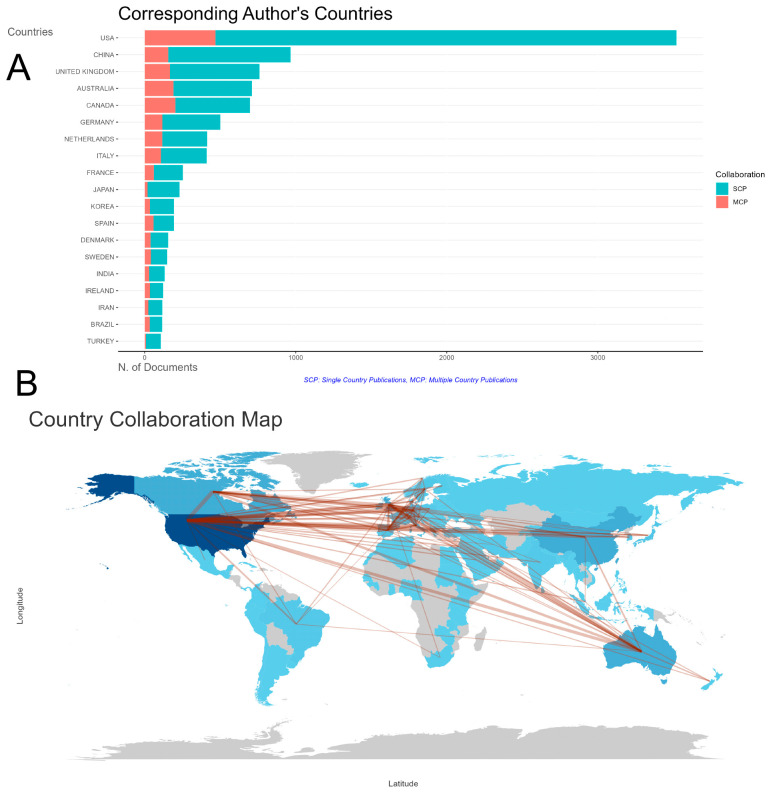
Publication situation of high-yield countries. The top 20 countries in terms of publication volume (**A**); cooperation relationships among countries (**B**).

**Figure 4 healthcare-12-01975-f004:**
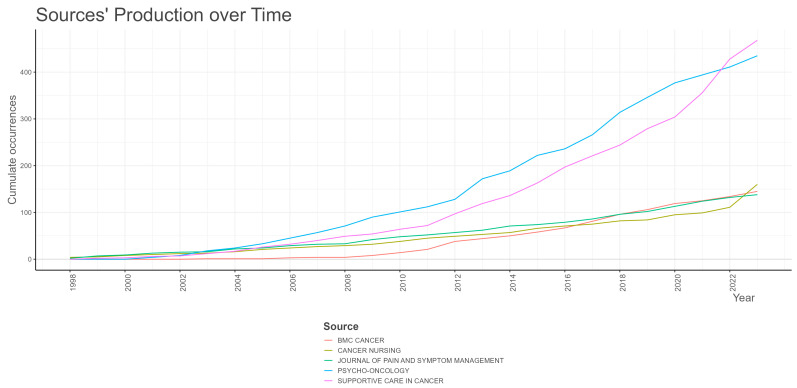
The publication volume of high-yield journals over the years.

**Figure 5 healthcare-12-01975-f005:**
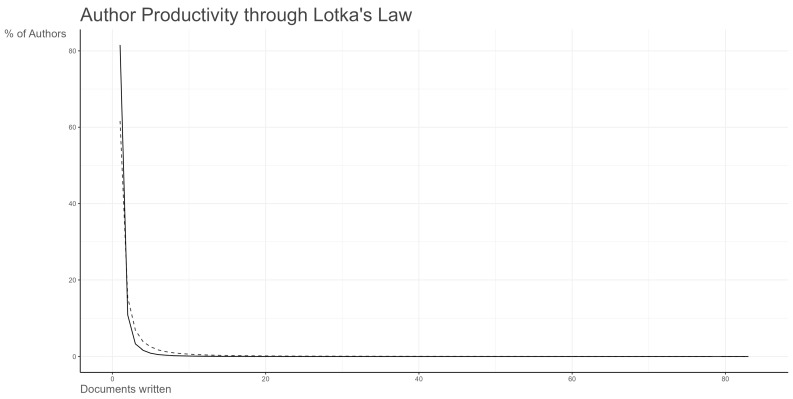
Analysis of core authors through Lotka’s Law. Solid line represents the actual trend line of the distribution of core authors in this field. Dotted line represents the predicted trend line of the distribution of core authors in this field.

**Figure 6 healthcare-12-01975-f006:**
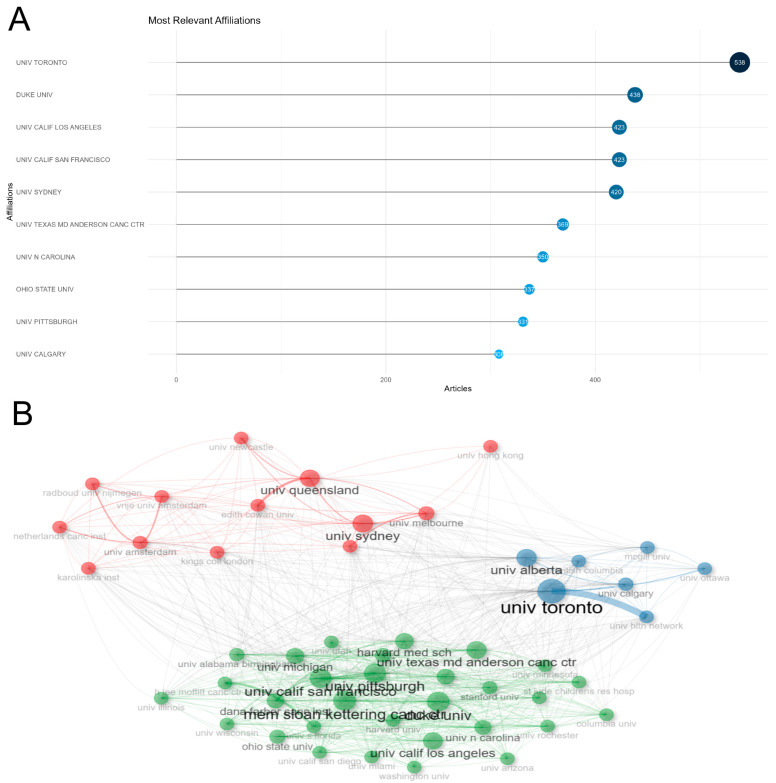
Publication situation of high-yield institutions. Top 10 institutions in terms of publication volume (**A**); cooperation relationships among institutions (**B**).

**Figure 7 healthcare-12-01975-f007:**
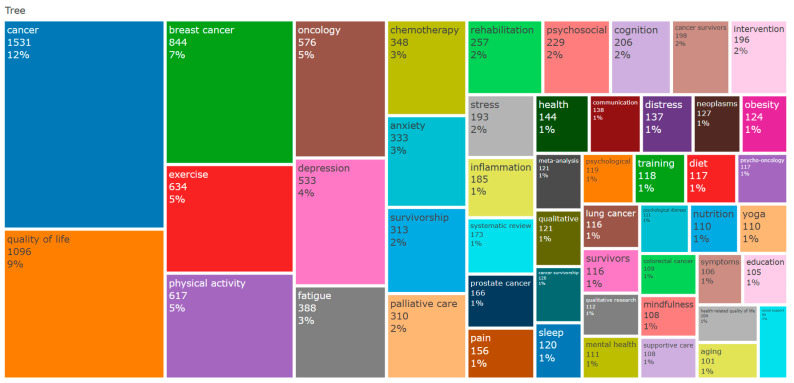
Dendrogram—Keyword frequency distribution.

**Figure 8 healthcare-12-01975-f008:**
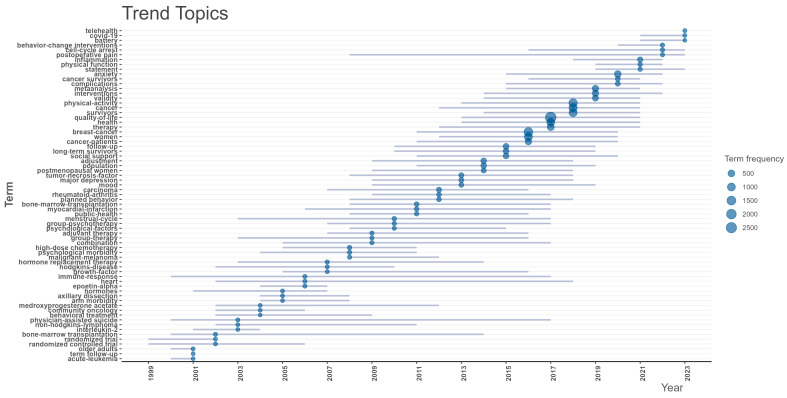
Keyword hotspot trend chart.

**Figure 9 healthcare-12-01975-f009:**
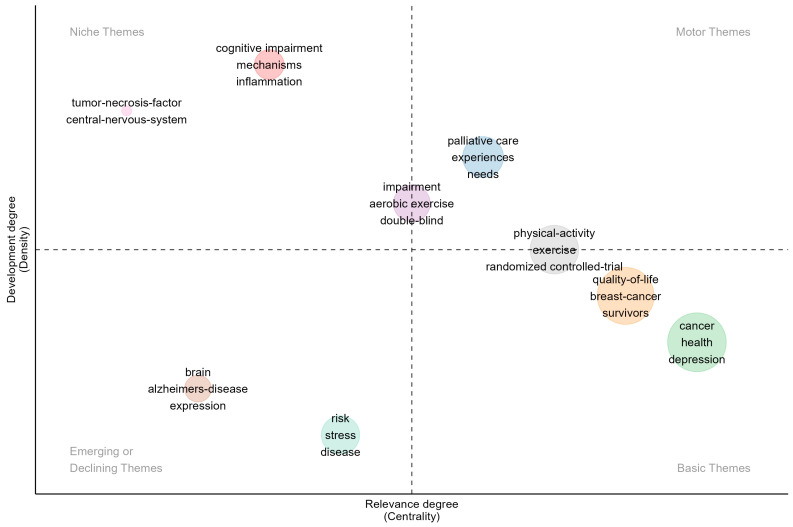
Theme evolution diagram.

**Table 1 healthcare-12-01975-t001:** Search strategy for Web of Science database.

Search Strategy	
#1	TS = (cancer* OR tumor* OR tumour* OR Neoplasm*)
#2	TS = (cogniti* OR brain*fog OR chemo*brain OR chemo*fog OR higher cerebral function OR neuropsychological OR psychologi* OR psychosocial)
#3	TS = (Physical activit* OR movement* OR exercis* OR train* OR Physical fitness OR Physical conditioning OR Cardiorespiratory fitness OR Endurance OR Muscle strengthening OR Weight lifting OR Weight bearing OR Stretching OR Qigong OR Run OR Swim* OR Sport* OR Athletic* OR Jog OR walk* OR biking OR bicycling OR cycling OR Yoga OR Pilates OR Tai Ji OR Tai Chi)
#4	#1 AND #2 AND #3

**Table 2 healthcare-12-01975-t002:** Top 10 countries and journals with the highest publication volume.

Ranking	Country	Articles	Citations	Journal	Articles	Citations
1	United States	3521	180,253	*Supportive Care in Cancer*	468	12,064
2	China	965	13,799	*Psycho-oncology*	435	14,935
3	United Kingdom	759	31,659	*Cancer Nursing*	160	2812
4	Australia	710	24,191	*BMC Cancer*	145	3691
5	Canada	697	25,926	*Journal of Pain and Symptom Management*	138	6520
6	Germany	501	18,796	*Cancer*	137	10,222
7	The Netherlands	414	16,892	*PLoS ONE*	134	3478
8	Italy	410	9252	*Journal of Clinical Oncology*	132	17,464
9	France	253	7529	*European Journal of Cancer Care*	131	2667
10	Japan	230	5226	*BMJ Open*	129	1219

**Table 3 healthcare-12-01975-t003:** Top 10 institutions by publication volume.

Ranking	Institution	Country	Articles
1	University of Toronto	Canada	538
2	Duke University	United States	438
3	University of California, Los Angeles	United States	423
4	University of California, San Francisco	United States	423
5	University of Sydney	Australia	420
6	The University of Texas MD Anderson Cancer Center	United States	369
7	University of North Carolina	United States	350
8	Ohio State University	United States	337
9	University of Pittsburgh	United States	538
10	University of Calgary	Canada	438

**Table 4 healthcare-12-01975-t004:** Top 10 most cited papers in the field of exercise intervention for CRCI.

First Author	Year	Title	Journal	Citations
Kerry S Courneya	2007	Effects of Aerobic and Resistance Exercise in Breast Cancer Patients Receiving Adjuvant Chemotherapy: A Multicenter Randomized Controlled Trial	*Journal of Clinical Oncology*	117
Julienne E Bower	2014	Cancer-related fatigue—mechanisms, risk factors, and treatments	*Nature Reviews Clinical Oncology*	107
Ruud Knols	2005	Physical Exercise in Cancer Patients During and After Medical Treatment: A Systematic Review of Randomized and Controlled Clinical Trials	*Journal of Clinical Oncology*	103
Karen M Mustian	2017	Comparison of Pharmaceutical, Psychological, and Exercise Treatments for Cancer-Related Fatigue A Meta-analysis	*JAMA Oncology*	97
Daniel A Galvão	2005	Review of Exercise Intervention Studies in Cancer Patients	*Journal of Clinical Oncology*	90
Bernardine M Pinto	2005	Home-Based Physical Activity Intervention for Breast Cancer Patients	*Journal of Clinical Oncology*	89
Maarten Hofman	2007	Cancer-Related Fatigue: The Scale of the Problem	*Oncology*	88
Maria Kangas	2008	Cancer-related fatigue: a systematic and meta-analytic review of non-pharmacological therapies for cancer patients	*Psychological Bulletin*	80
Hermann Faller	2013	Effects of Psycho-Oncologic Interventions on Emotional Distress and Quality of Life in Adult Patients With Cancer: Systematic Review and Meta-Analysis	*Journal of Clinical Oncology*	75
Linda E Carlson	2003	Mindfulness-Based Stress Reduction in Relation to Quality of Life, Mood, Symptoms of Stress, and Immune Parameters in Breast and Prostate Cancer Outpatients	*Psychosomatic Medicine*	71

## Data Availability

The data that support the findings of this study are available from the first author upon reasonable request.

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
