# Peer review of "Bibliometric Analysis of Research on Exercise Intervention for Cancer-Related Cognitive Impairments"

_healthcare, 2024, doi:10.3390/healthcare12191975_

Round 1
Reviewer 1 Report
Comments and Suggestions for Authors
Dear authors,
Thank you for your submission.
This is a well-written and informative manuscript. For the purpose of enhancing overall paper quality, I have made few suggestions for your consideration. Please make the following required changes:
Abstract
Results: please write what the abbreviations used stand for; The US leads ... please write United States (US)...
Keywords: it is recommended to arrange keywords in an alphabetical order.
1. Introduction
paragraph 2: exercise is important and the core of your paper! I suggest adding a sentence describes the long-term effect and the positive significant impact of exercise on overall physical and mental well-being of cancer patients.
2. Materials and Methods
2.2. Inclusion and exclusion criteria: in addition to the journals articles and language criteria, please add more specific information, such as topic of interest, year of publication, journals indexing and databases, etc.
Also, why did you use English language articles and rely solely on the WoSCC database? please elaborate as this could limit the external validity and generalizability of your study results.
4. Discussion
paragraph 3: and the UK ... please add United Kingdom (UK)
References
would it be possible to cite a more recent article instead of the Strang, 1998 one?
18. Strang, P. Cancer Pain--a Provoker of Emotional, Social and Existential Distress. Acta Oncol 1998, 37, 641–644, 511 doi:10.1080/028418698429973
Please make the required edits and modifications and I will be happy to review the revised version.
Many thanks.
Best wishes,
Reviewer 2 Report
Comments and Suggestions for Authors
1. Many bibliometrics studies face reproducibility problems and do not provide new valuable insights for the academic community. I hope the authors can avoid these problems.
2. Line 50-64: for the introduction of Bibliometrics, it is better to use authoritative studies from journals such as Scientometrics, Journal of Informetrics, and QSS.
3. Line 65-66: please use evidence to justify your choice of Web of Science rather than other databases such as Scopus.
4. By following the suggestions of the following study, the sub-datasets and corresponding coverage years of your subscribed Web of Science Core Collection should be given.
Liu, W. (2019). The data source of this study is Web of Science Core Collection? Not enough. Scientometrics, 121(3), 1815-1824.
5. Line 89: it is not precise to allocate “review” as original research.
6. Line 91: the used tool should be cited.
7. Line 116-117: it is strange to find the first publication in 1998. Possible explanations should be given. The caveats for the use of Web of Science Core Collection in old literature retrieval and historical bibliometric analysis as uncovered in previous research should be one possible reason.
8. Figure 1 is not clear.
9. Figure 2 is not clear.
10. Line 187-190: Please explain how the problem of name ambiguity was resolved.
11. Figure 4: what does the keyword “of-life” mean?
12. Figure 5: it is not informative.
13. Line 392-393: please also mention China’s remarkable role in the production of scientific articles, especially for SCI-index articles in recent years.
14. The limitations of Web of Science as documented by many studies from the Scientometrics community should be mentioned precisely.
Comments on the Quality of English Languageshould be polished
Reviewer 3 Report
Comments and Suggestions for Authors
Dear Author's
Thank you for the opportunity to read the study results. I actually have two comments/suggestions: 1. in abstract please replace Discussion on Results 2. please attach a Prisma diagram - it will definitely facilitate the analysis of the selection of research materialbest regards
Reviewer 4 Report
Comments and Suggestions for Authors
The paper "Bibliometric Analysis of Research on Exercise Intervention for Cancer-Related Cognitive Impairments" examines the impact of exercise interventions on cognitive issues caused by cancer treatments through an analysis of 10,345 articles. It highlights trends in publications, key countries, researchers, and institutions in detail. Current research focuses on exercise interventions for cognitive rehabilitation, especially in breast cancer patients. The study also acknowledges progress over the past 20 years but notes ongoing challenges.
General comment: the study is well-written and well-designed and can be accepted after have addressed the following suggestions:
-
Figures: Please, improve the quality and size of figures for better readability.
-
Discussion: The discussion section should contain future directions and provide more critical insights about the data, instead of just summarizing them. For instance, the part related to technological advancements could be stressed. New technologies, such as telemedicine and virtual reality, are gaining a lot of attention as tools for improving the quality of life and for rehabilitation programs. Studies on usability and feasibility should also be highlighted. Does the data confirm this hypothesis of a growing trend? If yes, why in your opinion?
Similarly, the growing interest in the effects of exercise interventions on the quality of life in specific cancer types and groups should be emphasized and stressed with examples and critical insights.
-
Limits and Strengths: Place the study's limits in a separate section titled "Limits and Strengths of the Study."
Overall, the paper shows promise, but addressing the aforementioned recommendations will greatly augment its clarity, rigor, and relevance of the findings.
Round 2
Reviewer 2 Report
Comments and Suggestions for Authors
Some of my concerns were addressed, but the manuscript still needs to be revised before it can be accepted for publication.
1. Line 18: what does “leads (US)” mean?
2. Line 82-83: the description about “Web of Science” is not precise. Please refer to studies about “Web of Science” published in Scientometrics or Quantitative Science Studies in recent years.
3. Line 84-85: the phrase “most comprehensive and reliable” is not precise. Please refer to authoritative studies published in Scientometrics or Quantitative Science Studies.
4. Line 87-90: It is good to specify the sub-datasets and coverage years of the Web of Science Core Collection to readers. It is better to give more explanations of this important information to readers and acknowledge the suggestions from previous studies.
5. Line 87-90: please delete the confusing date “March 12, 2024” and add the access date of your data.
6. Line 116: the word “irrelevant” is not precise.
7. Line 123: please refer to the study “Caveats for the use of Web of Science Core Collection in old literature retrieval and historical bibliometric analysis” and explain this as a limitation of your study.
8. Line 132-141: the citation indicator should be explained more clearly.
9. Figure 2: the growth of publications may also be due to the inclusion of new sub-datasets and the expansion of Web of Science. You can refer to related studies for more information. These points should be disclosed to readers. For example, the following study is about the inclusion of new sub-datasets.
Liu, F. (2023). Retrieval strategy and possible explanations for the abnormal growth of research publications: Re-evaluating a bibliometric analysis of climate change. Scientometrics, 128(1), 853-859.
10. Table 3: the country information should be added.
11. Line 409-412: please add authoritative references focusing on “Web of Science” or “Scopus” to support these points.
Comments on the Quality of English Languageshould be polished
